# Autoantibodies from Patients with Scleroderma Renal Crisis Promote PAR-1 Receptor Activation and IL-6 Production in Endothelial Cells

**DOI:** 10.3390/ijms222111793

**Published:** 2021-10-30

**Authors:** Michèle Simon, Christian Lücht, Isa Hosp, Hongfan Zhao, Dashan Wu, Harald Heidecke, Janusz Witowski, Klemens Budde, Gabriela Riemekasten, Rusan Catar

**Affiliations:** 1Department of Nephrology and Internal Intensive Care Medicine, Charité—Universitätsmedizin Berlin, Corporate Member of Freie Universität Berlin and Humboldt-Universität zu Berlin, 10117 Berlin, Germany; michele.simon@charite.de (M.S.); christian.luecht@charite.de (C.L.); isa@lutzschramm.de (I.H.); hongfan.zhao@charite.de (H.Z.); Dashan.wu@charite.de (D.W.); jwitow@ump.edu.pl (J.W.); klemens.budde@charite.de (K.B.); 2CellTrend GmbH, 14943 Luckenwalde, Germany; heidecke@celltrend.de; 3Department of Pathophysiology, Poznan University of Medical Sciences, 60-806 Poznan, Poland; 4Clinic for Rheumatology and Clinical Immunology, Universitätsklinikum Schleswig-Holstein, 23538 Lübeck, Germany; Gabriela.Riemekasten@uksh.de

**Keywords:** systemic sclerosis, scleroderma renal crisis, PAR-1 receptor, IL-6, autoantibodies

## Abstract

Background. Scleroderma renal crisis (SRC) is a life-threatening complication of systemic sclerosis (SSc). Autoantibodies (Abs) against endothelial cell antigens have been implicated in SSc and SRC. However, their detailed roles remain poorly defined. Pro-inflammatory cytokine interleukin-6 (IL-6) has been found to be increased in SSc, but its role in SRC is unclear. Here, we aimed to determine how the autoantibodies from patients with SSc and SRC affect IL-6 secretion by micro-vascular endothelial cells (HMECs). Methods. Serum IgG fractions were isolated from either SSc patients with SRC (n = 4) or healthy individuals (n = 4) and then each experiment with HMECs was performed with SSc-IgG from a separate patient or separate healthy control. IL-6 expression and release by HMECs was assessed by quantitative reverse transcription and quantitative PCR (RT-qPCR) and immunoassays, respectively. The mechanisms underlying the production of IL-6 were analyzed by transient HMEC transfections with IL-6 promoter constructs, electrophoretic mobility shift assays, Western blots and flow cytometry. Results. Exposure of HMECs to IgG from SSc patients, but not from healthy controls, resulted in a time- and dose-dependent increase in IL-6 secretion, which was associated with increased AKT, p70S6K, and ERK1/2 signalling, as well as increased c-FOS/AP-1 transcriptional activity. All these effects could be reduced by the blockade of the endothelial PAR-1 receptor and/or c-FOS/AP-1silencing. Conclusions. Autoantibodies against PAR-1 found in patients with SSc and SRC induce IL-6 production by endothelial cells through signalling pathways controlled by the AP-1 transcription factor. These observations offer a greater understanding of adverse endothelial cell responses to autoantibodies present in patients with SRC.

## 1. Introduction

Scleroderma renal crisis (SRC) is a rare, but potentially life-threatening complication of systemic sclerosis (SSc) [1]. It may occur in up to 13% of all SSc patients [2,3,4,5] with those with early (<5 years) or rapidly progessing diffuse cutaneous SSc (dcSSc) being predominantly affected [6]. Typically, clinical symptoms of SRC are associated with a rise in blood pressure and serum creatinine [6]. In addition, there may be symptoms of microangiopathic hemolysis or thrombosis. As a result, SRC was until recently the leading cause of death in SSc [7]. However, the introduction of angiotensin-converting enzyme inhibitors (ACEi) markedly improved the prognosis in these patients [7]. Nevertheless, renal involvement is still a severe, but nowadays often more silent complication of the disease, and as such determines the patient survival [8]. So far, studies on the mechanisms of SRC are scarce.

The formation of autoantibodies is a cardinal feature of systemic autoimmune diseases [9]. The presence of SSc autoantibodies is associated with kidney fibrosis and vasculopathy during SRC [10]. Some of these antibodies target endothelial antigens and therefore, the vascular endothelium forms a critical interphase between parenchymal cells and the immune system. Indeed, antibody-induced endothelial dysfunction leads to the induction of pro-inflammatory cytokines, perivascular infiltration of immune cells, and vascular remodeling, contributing to SRC [11]. As serum levels of IL-6 are increased in SSc [12], it has been hypothesized that IL-6 may serve as a biomarker to stratify patients for SRC risk and response to therapy [13]. However, for this purpose, it would be essential to elucidate the mechanisms involved in the regulation of IL-6 secretion in SRC.

In recent years, a new group of autoantibodies that react with cell surface receptors like G-Protein-coupled receptors (GPCRs), and contribute to the pathological activation of intracellular signalling has been identified [14]. Anti-GPCR autoantibodies against endothelial angiotensin II type 1 receptor (AT1-R) and endothelin-1 type A receptor (ETA-R) are commonly detected in SSc [15], suggesting their involvement in the SSc pathogenesis [11]. In contrast, the role of autoantibodies against Protease-activated receptor-1 (PAR-1), another GPCR, is less clear. It is known that thrombin-activated PAR-1 can contribute to inflammation [16,17,18], which may include IL-6 production [19]. Thus, the objective of the present study was to analyze the mechanisms by which the IgG fraction found in SSc patients modulates signalling through PAR-1 and affects IL-6 secretion by human endothelial cells (ECs) (Figure 1).

## 2. Results

### 2.1. Effect of SSc-IgG on IL-6 Release in HMECs

Demograhic data, clinical manifestation, histological findings and antibo6dy levels for each SSc patient with SRC included are shown in Appendix A. Exposure of HMECs to the IgG fraction of serum from SSc patients (SSc-IgG) resulted in a time- and dose-dependent increase in IL-6 release. The effect of SSc-IgG at 1 mg/mL became apparent after 6 h of stimulation, peaked at 12 h and plateaued at 24 h (Figure 2A). Such an effect was also dose-dependent and reached maximum at the highest dose of SSc-IgG tested (2 mg/mL; Figure 2B). In contrast, the IgG fraction of serum from patients with rheumatoid arthritis (RA), a chronic systemic disease with immunologically mediated inflammation [20] had no effect on IL-6 production by HMECs (Appendix A). 

Since antibodies against AT1-R and ETA-R have been linked to the pathogenesis of SSc [11], we analyzed whether signalling through AT1-R and ETA-R mediated the effect of SSc-IgG on IL-6 release. To this end, HMECs were preincubated with either AT1-R or ETA-R inhibitors (valsartan or sitaxentan, respectively) at pharmacologically relevant doses [21]. The use of these inhibitors had no effect on IL-6 secretion induced by SSc-IgG (Figure 2C). In contrast, pretreatment of HMECs with BMS-200261, a PAR-1 inhibitor, abolished the stimulating effect of SSc-IgG on IL-6 release (Figure 2D). BMS-200261 was selected after preliminary experiments, which showed that BMS-200261 had alone no effect on IL-6 release (Appendix A). PAR-1 functionality was confirmed by stimulating HMECs with thrombin, which is known to signal through PAR-1 [18]. These experiments showed that thrombin induced IL-6 release by HMECs and that this effect was reduced when HMECs were pretreated with BMS-200261 (Appendix A).

### 2.2. SSc-lgG-Induced PAR-1 Activation

Next, we assessed the expression of PAR-1 on the surface of HMECs. Expression of activated PAR-1 on unstimulated HMECs or on HMECs stimulated with IgG from healthy controls (Con-IgG) was less than 10% of total PAR-1 expression (Figure 3A). In contrast, expression of activated PAR-1 on HMECs treated with SSc-IgG (1 mg/mL) was >80%, which was similar to PAR-1 activation exerted by thrombin (Figure 3A). The effect of SSc-IgG and thrombin was almost completely eliminated by the PAR-1 Inhibitor BMS-200261 (Figure 3A). In contrast, PAR-1 activation by SSc-IgG was not reduced by Refludan, a thrombin inhibitor (Figure 3B), which however abolished PAR-1 activation by thrombin (Appendix A). These effects suggested that SSc-IgG activates PAR-1 through a mechanism different from that induced by thrombin. 

### 2.3. PAR-1 Blockade Impairs SSc-IgG-Induced Intracellular Signalling in HMECs

Since thrombin signalling through PAR-1 has previously been shown to involve phosphatidylinositol 3-kinase (PI3K), as mammalian target of rapamycin (mTOR) and the extracellular signal-regulated kinases 1/2 (ERK1/2) [22], we examined whether SSc-IgG could initiate similar signalling pathways in HMECs. Indeed, exposure of HMECs to SSc-IgG at a dose of 1 mg/mL resulted in an approximately 2-fold increase in the expression of phosphorylated pAKT (a central effector of the PI3K pathway upstream of mTORC1), p70S6K (a substrate for the mTOR pathway) and pERK1/2 (Figure 4). Importantly, such effects were not detected in HMECs stimulated with IgG from healthy donors and were almost entirely abolished by preincubation of HMECs with the PAR-1 inhibitor, BMS-200261 (Figure 4). 

### 2.4. SSc-IgG Induces ERK1/2-Mediated IL-6 Release by HMECs

We next analyzed whether activation of ERK1/2 affects the release of IL-6 by HMECs. Pretreatment of HMECs with PD-184352, a specific ERK1/2 inhibitor, resulted in a dose-dependent decrease in IL-6 release, with the significant effect observed at concentrations ≥ 0.1 μM (Figure 5). 

### 2.5. SSc-IgG Activated the IL-6 Promoter in Microvascular Endothelial Cells

To investigate how SSc-IgG affects the activity of the IL-6 gene promoter, we employed a set of established assays [23,24]. First, HMECs were transiently transfected with IL-6 luciferase reporter gene constructs and stimulated with SSc-IgG (1 mg/mL). Exposure of HMECs to SSc-IgG led to a significant increase in the full-length IL-6 promoter activity (Figure 6A). To identify IL-6 promoter regions responsive to SSc-IgG, progressive deletions of the IL-6 promoter were performed. Truncation of the promoter region spanning positions −160 to +11 abolished the ability of the IL-6 promoter to respond to SSc-IgG (Figure 6A), suggesting that the region contained regulatory elements essential for the IL-6 promoter activity. The in silico analysis pointed to the presence of high-affinity binding sites for c-FOS. To determine whether c-FOS mediated the effect of SSc-IgG on the IL-6 promoter, an electrophoretic mobility shift assay (EMSA) was performed using a biotin-labeled consensus oligonucleotide for c-FOS binding that corresponded to positions −278 to −257 of the IL-6 promoter (Figure 6B). This experiment demonstrated that nuclear extracts from cells stimulated with SSc-IgG formed a DNA–protein complex with the specific oligonucleotide and that the effect was less in the presence of competing unlabeled DNA. 

We then examined the binding specificity of AP-1/c-FOS by analyzing the effect of the mutation in the AP-1 binding site on the activity of the IL-6 promoter. In contrast to cells transfected with a normal construct, transfection with a construct mutated within the AP-1 binding site abolished IL-6 promoter activation after stimulation with SSc-IgG (Figure 6C) indicating that functional binding of active AP-1 plays a crucial role for IL-6 promoter activation and subsequent IL-6 release. 

Since c-FOS is part of the AP-1 transcription complex, we also assessed the effect of SSc-IgG on c-FOS mRNA expression. Exposure of HMECs to SSc-IgG resulted in an approximately 5-fold increase in c-FOS mRNA expression and this effect was significantly diminished by pretreatment of the cells with the PAR-1 blocker, BMS-200261 (Figure 6D). The contribution of AP-1/c-FOS signalling to the SSc-IgG-driven IL-6 release was confirmed by the observation that the effect of SSc-IgG was significantly reduced by SR-11302, a specific AP-1 inhibitor (Figure 6E) and by silencing c-FOS with specific siRNA (Figure 6F). 

## 3. Discussion

The main observation of the present study is that antibodies from SSc patients with SRC, but not from healthy individuals, are capable of inducing IL-6 production in endothelial cells, acting through the PAR-1. Intriguingly, this effect of SSc-IgG on PAR-1 seems to resemble that of thrombin [17] and engages downstream signalling through the PI3K/mTOR/ERK1/2 pathway and AP-1/c-FOS transcription factor.

The exact role of IL-6 in SSc and SRC is not well understood. Pro-inflammatory functions of IL-6 in SSc have been recognized in several studies [12,25,26,27]. Surprisingly, however, no obvious signs of inflammation were found in kidney biopsies from SRC patients [28]. To the best of our knowledge, there is only one clinical case report that suggests an association between SRC and IL-6. It documents improvement in creatinine clearance following therapy with tocilizumab, an anti-IL-6 receptor antibody, in a patient with renal failure due to SRC [29]. The lack of apparent inflammatory lesions in the kidneys during SRC may be related to the fact that pro-inflammatory effects of IL-6 are seen only during a specific stage of SSc [30] and serum IL-6 levels decrease gradually with disease duration [31]. Nevertheless, SRC is present in patients with early and rapidly progressing disease with an inflammatory phenotype. Our data indicate a potential role of IL-6-targeted therapies also in SRC. In addition, addressing signalling of SSc-IgG, targeting the AP-1/c-FOS transcription factor complex could be another therapeutic option. 

Functional antibodies against AT1-R and ETA-R are detected commonly in SSc with higher levels associated with a more severe course of disease and increased mortality [15]. They were found to induce interleukin-8 (IL-8) and vascular cell adhesion molecule-1 (VCAM-1) expression in HMECs in vitro and in naive C57BL/6J mice in vivo [11]. Importantly, ACE-inhibitors are known to improve endothelial cell function and their introduction into clinical practice led to a significant decrease in SRC-related mortality [28]. Moreover, the blockade of ETA-R signalling was shown to reduce IL-6 expression in a rat model of chronic kidney disease [32]. However, our findings do not support the idea that an increase in IL-6 secretion by HMECs stimulated with SSc-IgG is mediated by functional anti-AT1-R and/or ETA-R Abs, since the use of specific inhibitors of AT1-R and ETA-R didn’t reduce the secretion of IL-6 in vitro. In contrast, we have demonstrated that this effect is mediated rather through PAR-1 signalling (Figure 2). These data show that in addition to AT1-R/ETA-R Abs, PAR-1 Abs contribute to the pathogenesis of SSc, particularly to that of SRC. Thus, PAR-1 inhibition could be a specific therapeutic option in the treatment of SRC. 

The role of PAR-1 as a key receptor for thrombin is recognized [33,34]. Here, we demonstrate that SSc-IgG acts in a thrombin-like fashion to activate PAR-1. First, the effect of SSc-IgG on PAR-1 activation can be blocked by the thrombin inhibitor refludan. Second, the signalling pathway evoked by SSc-IgG resembles closely that induced by thrombin on endothelial VEGF expression [35]. 

Studies on the function of autoantibodies against PAR-1 are rare, with only one report [36] of clinical relevance published. That analysis of 197 women with primary epithelial ovarian cancer revealed a negative correlation between the levels of anti-PAR-1 antibodies and histological grading of the tumor. 

This finding is in line with our observation that AKTSer473, which is an effector of the PI3K-pathway involved in endothelial proliferation, survival and angiogenesis [37], can be affected by anti-PAR-1 Abs in SSc-sera. A stimulatory effect of SSc-IgG on AKTSer473 was also found in vascular smooth muscle cells (VSMCs) [38]. In that study antibodies against the platelet-derived growth factor receptor (PDGFR) were found to transactivate the epidermal growth factor receptor (EGFR) leading to phosphorylation of AKT at Ser473 and ERK1/2 activation, resulting in increased protein synthesis, as well as increased expression of pro-fibrotic genes. Another study [39] demonstrated that anti-fibroblast antibodies found in patients with SSc stimulated fibroblasts to produce pro-fibrotic and pro-angiogenic chemokines through a mechanism controlled by ERK1/2. Involvement of ERK1/2 signalling has also been implicated in renal fibrosis associated with systemic lupus erythematosus [40]. In addition, ERK1/2 signalling was demonstrated to underlie IL-6 production in human primary mesangial and proximal tubular cells [41]. Sustained activation of ERK1/2 was also found to be involved in renal endothelial inflammation [42], myofibroblast differentiation, extracellular matrix production [43] and fibrosis [44] in the course of SSc. Interplay between PAR-1 and p70S6K was also demonstrated in animal models of neuronal ischemia [45] and endometriosis [46]. Corroborating our results, phosphorylation of p70S6K was also detected in the renal endothelium of patients with SRC [47]. 

Organs affected by autoimmune disorders are infiltrated by immune cells that crosstalk with resident stromal cells, e.g., fibroblasts or endothelial cells to augment and perpetuate the inflammatory response and tissue remodeling [48]. In this respect, by connecting metabolic cues with inflammatory cytokines, mTOR may be another promising therapeutic target in autoimmune disorders [49]. Though, in the present manuscript, we have identified a role of IL-6 in SRC independent of inflammation. 

As a member of the FOS multigene family, c-Fos heterodimerizes with other components to form the AP-1 transcription factor [50]. Being a crucial integrator of a myriad of extracellular signals, AP-1 switches on/off different transcriptional programs depending on the nature of the signal, the cellular context and/or the combination of the dimer components involved [50]. A prominent role of AP-1 in SSc-associated tissue fibrosis has been elucidated before. In this respect, pathological activation of fibroblasts resulting in the accumulation of extracellular matrix (ECM) was demonstrated to involve the AP-1 components: fos-related antigen-2 (Fra-2) [51] from the Fos multigene family [50], and JunD from the Jun multigene family [52]. Moreover, the AP-1 constituents c-Jun and c-Fos were implicated in generating a pro-fibrotic tissue milieu by inducting a pro-fibrotic and pro-inflammatory secretory phenotype in monocytes [53]. Additionally, our data are in line with animal studies showing severe diffuse SSc in mice transgenic for Fra-2 (Fra-2 tg) [54]. With extensive changes in ECM blood vessels, Fra-2 tg mice are considered a promising preclinical model to study the complex interaction between vasculopathy and fibrosis in SSc [55]. These studies revealed increased perivascular inflammation in the skin of Fra-2 tg mice [55] and increased systemic concentration of IL-6 in Fra-2 tg mice exposed to pulmonary infection. In accordance, the osteoblast-specific overexpression of Fra-2 resulted in increased systemic levels of IL-6 [56]. 

Among other effects, c-FOS has also been implicated in pro-inflammatory cytokine-induced IL-6 expression in vascular cells [23]. Additionally, more recently, c-FOS has been shown to mediate hyper-inflammatory endothelial IL-6 trans-signalling in response to the SARS-CoV-2 spike protein [57]. Other in vitro studies showed that the natural PAR-1 ligand, thrombin, provoked a time- and concentration-dependent IL-6 release in endothelial cells [58]. The application of hirudin, a natural thrombin inhibitor, resulted in a reduced expression of both PAR-1 and IL-6, as well as decreased accumulation of ECM in the renal interstitium [59]. 

Although the advent of ACEi reduced mortality in SRC patients, 1-year-outcomes remain poor [28]. This may indicate that novel treatments are required to improve the prognosis in SRC patients [28]. These may be helped by detailed characterization of interactions between SSc-associated autoantibodies and vascular endothelial cells. We here show, that PAR-1 Abs contribute to pathological endothelial processes in SSc, especially in SRC by signalling through PI3K/mTOR/ERK1/2 and AP-1/c-FOS resulting in an IL-6 release, for the first time. Thus, these results are of more decisive clinical significance. Hirudin-analogs [60] and Voraxapar, an orally applicable PAR-1 inhibitor, are already in clinical use [61]. PD-184352, the ERK1/2 inhibitor, was successfully applied in phase 1 studies as orally available compound Cl-1040 [62]. For inhibition of mTOR, Everolimus is an orally administered, accessible approved drug [63]. Tocilizumab, an anti Il-6 receptor antibody, appears to demonstrate beneficial effects in a phase 3 study of systemic sclerosis-associated interstitial lung disease (SSc-ILD) [64]. Small molecule inhibitors of AP-1 have also been proposed as an attractive therapeutic strategy [65,66]. For example, T-5224, a small-molecule inhibitor suppressing the DNA-binding activity of the dimer c-Jun/c-Fos revealed an excellent safety profile in a preclinical animal study of arthritis [67]. Therefore, based on our preclinical results, a therapy applying one or more of the aforementioned accessible therapeutic agents appears reasonable to treat SRC in SSc patients.

## 4. Materials and Methods

### 4.1. Materials

Unless stated otherwise, all chemicals were from Sigma-Aldrich (St Louis, MO, USA) and all culture plastics were Falcon from Becton Dickinson (Franklin Lakes, NJ, USA). Cell culture media and buffers were from Thermo Fisher Scientific (Waltham, MA, USA) and fetal calf serum (FCS) was from Invitrogen (Darmstadt, Germany). The source and characteristics of antibodies used is given in Appendix A. Thrombin used was α-thrombin from human plasma with concentrations reported in NIH units of activity.

### 4.2. Endothelial Cell Culture

Human dermal microvascular endothelial cells (HMECs, catalogue no. CRL-3243) were purchased from ATCC^®^ (Manassas, VA, USA) and cultured as described previously [35]. 

### 4.3. Patient Samples

SSc patients were those who were treated for SRC at the Charité University Hospital Campus Mitte in Berlin from 1 January 2010, through 31 July 2014. Samples of serum were obtained from all patients and screened for agonistic antibodies targeting AT1-R, ETA-R and PAR-1, as described previously [15]. Control samples were obtained from healthy subjects. Written informed consent to use serum samples for research purposes was obtained from each participant. The study was approved by the institutional review board of the Charité Universitätsmedizin Berlin (AZ/Nr.: EA2/068/07). 

### 4.4. IgG Isolation

IgG was isolated by protein-G sepharose chromatography in 20 mM phosphate buffer pH 7.0 and eluted with 0.1 M glycine/HCl, pH 2.7; then the pH was neutralized with 1 M Tris/HCl, pH 9.0 and eluted IgG was dialyzed against HMEC medium MCDB131. Absorbance of the solution obtained was measured at 280 nm (Emax, Molecular Devices, San Jose, CA, USA) [11].

### 4.5. IL-6 Protein Measurement

IL-6 protein concentration was measured by immunoassay using a human IL-6 Antibody Pair Kit (Thermo Fisher Scientific, Waltham, MA, USA), as previously described [68]. The limit of detection was 2 pg/mL as indicated in the product specification.

### 4.6. Gene Expression Analysis

Expression of the *IL-6* gene as well as *β2M* as a housekeeping gene was assessed with reverse transcription and quantitative PCR (RT-qPCR) as detailed elsewhere [23]. 

### 4.7. DNA Construct Transfection and Luciferase Assays

Genomic DNA from HMECs was isolated with the Isol-RNA Lysis Reagent (5Prime, VWR, Radnor, PA, USA) and used to generate progressive IL-6 5′-deletion luciferase plasmid constructs (pLuc 2211, pLuc 1211, pLuc 611 and pLuc 171) via PCR amplification using appropriate primer pairs. The Infusion Cloning Kit (Clontech; Takara Bio USA, Mojntain View, CA, USA) was used with the pGL4.10 vector backbone to create the luciferase reporter constructs as described elsewhere [69]. The length of the promoter segments was checked for correctness by restriction digestion and by sequencing (LC Genomics, Houston, TX, USA). 

### 4.8. Transient Transfection and Luciferase Assays

For transient transfection studies cells were seeded into 6-well culture plates at a density that allowed them to reach 70–80% confluence within 24 h. Transfections were then performed in the absence of serum using TurboFect™ transfection reagent (Thermo Fisher Scientific, Waltham, MA, USA) at a ratio of 1 mL/0.33 mg according to the manufacturer’s instructions and as detailed previously [69]. HMECs were transfected with the *IL-6* reporter plasmid (0.2 lg/well) and co-transfected with the reference pRL-TK Renilla plasmid (0.02 lg/well). Luciferase activity was assessed with the dual-luciferase reporter assay system (Promega, Mannheim, Germany) with a microplate luminometer (Fluostar Optima; BMG Labtech, Ortenberg, Germany) and normalized to background levels of Renilla luciferase activity from the cotransfected control vectors. Transfection of HMECs with either siRNA for *c-FOS* (sc-29221) or scrambled siRNA control (sc-37007) was performed with the siRNA Transfection Reagent (all from Santa Cruz Biotechnology, Dallas, TX, USA). The region -599 to -161 of the human *IL-6* promoter (GenBank NC_000007.14) was analyzed with Transcription Element Search Software (http://www.cbil.upenn.edu/cgi-bin/tess/tess, accessed on 11 April 2021) to predict and locate potential transcription factor binding sites.

### 4.9. Nuclear Extracts and Electrophoretic Mobility Shift Assay

Nuclear extracts were prepared using NE-PER Nuclear and Cytoplasmic Extraction Kit and oligonucleotide probes labeled with Biotin 3′-End DNA Labeling Kit (both from ThermoFisher Scientific, Waltham, MA, USA). For electrophoretic mobility shift assay (EMSA) [68], the following *AP-1* probe was used (the corresponding region of the *IL-6* promoter is given in parenthesis): 5′-CAAAGTGCTGAGTCACTAATAA-3′ (−278 to −257). Each binding mixture (20 µL) contained 5 µg of nuclear extract, 20 fmol labeled double-stranded probe, 1 µg poly-dI/dC and 2 µL 10× reaction buffer and was incubated at room temperature for 30 min. In supershift experiments nuclear extracts were incubated with SSc-IgG for 20 min at room temperature before addition of the biotin-labeled probe. Protein–DNA complexes then were analyzed by electrophoresis in 6% non-denaturing polyacrylamide gels and visualized using the LightShift Chemiluminescent EMSA Kit (ThermoFisher Scientific, Waltham, MA, USA).

### 4.10. Western Blotting

Preparation of cell extracts was conducted as described before [23], electrophoresed on sodium dodecyl sulfate-polyacrylamide gels and Western blotted using antibodies against pAKT, p70S6K, pERK1/2 and α-Tubulin (Cell Signaling Technology, Frankfurt, Germany) as well as appropriate secondary peroxidase-conjugated IgG (Dianova, Hamburg, Germany). The bands obtained were visualized and analyzed using Enhanced Chemiluminescence Detection System (ThermoFisher Scientific, Waltham, MA, USA) and Image J 1.43 software (National Institutes of Health, Bethesda, MD, USA).

### 4.11. PAR-1 Expression and Activation

PAR-1 receptor expression and cleavage was assessed by flow cytometry (FACS Aria; Becton Dickinson, Franklin Lakes, NJ, USA). Activation of PAR-1 was monitored using SPAN12, a monoclonal antibody detecting an epitope of PAR-1 that exists only in the uncleaved, i.e., non-active PAR-1. Thus, loss of SPAN12 staining points to PAR-1 receptor activation [70]. To prevent unspecific PAR-1 activation and internalization all experimental steps were performed at 4 °C. HMECs were diluted (3:7) in PBS and incubated with either SPAN12 or a buffer for 10 min, and then fixed with 1% paraformaldehyde and analyzed immediately by flow cytometry. 

### 4.12. Statistics

Statistical analysis was performed using GraphPad Prism 6.05 software (GraphPad Software, San Diego, CA, USA). The data were analyzed by t-test or repeated measures analysis of variance, as appropriate. Results were expressed as means ± SEM. Differences with a *p*-value <0.05 were considered significant. Asterisks represent *p* values as follows: * for *p* < 0.05, ** for *p* < 0.01, and *** for *p* < 0.001.

## Figures and Tables

**Figure 1 ijms-22-11793-f001:**
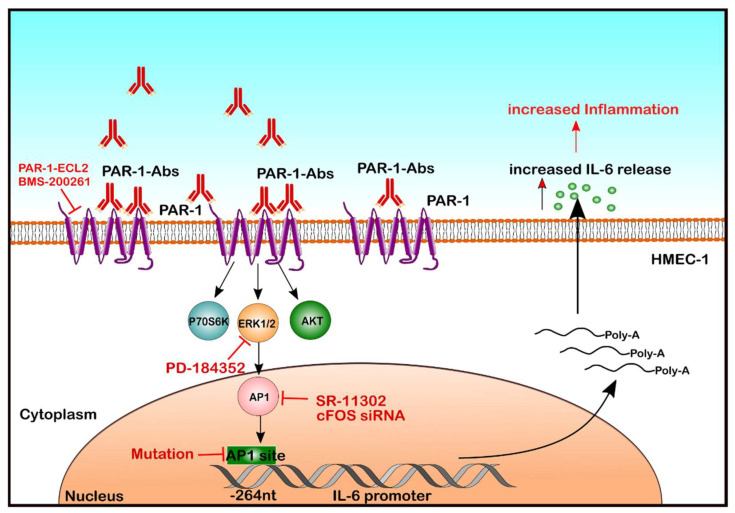
Molecular mechanisms underlying endothelial IL-6 release in response to autoantibodies from patients with SRC. Here, we demonstrate that autoantibodies present in SRC patients can activate GPCRs on EC and engage a signalling cascade involving PAR-1, PI3K/mTOR/ERK1/2 and AP-1/c-FOS to activate the IL-6 promoter (Position @-264 nucleotides from the IL-6 gene), which leads to IL-6 mRNA expression and IL-6 protein release.

**Figure 2 ijms-22-11793-f002:**
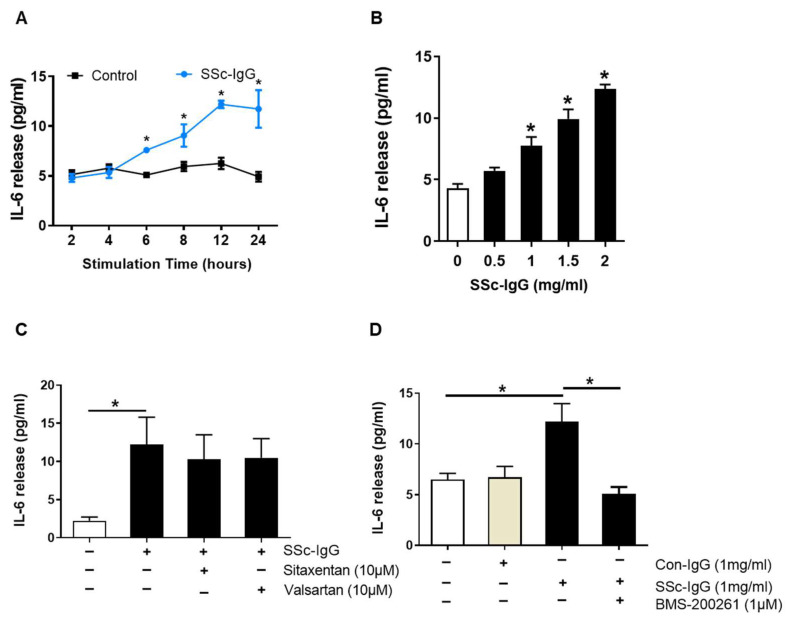
Effect of SSc-IgG on IL-6 production by human microvascular endothelial cells. HMECs were cultured in the presence or absence of SSc-IgG and assessed for (**A**) time- and (**B**) dose-dependent IL-6 secretion (n = 4). The exposure time in (**B**) was 24 h, while the dose of SSc-IgG in (**A**) was 1.0 mg/mL. In separate experiments, HMECs were pretreated for 1 h with (**C**) either valsartan (AT1-R inhibitor) or sitaxentan (ETA-R inhibitor) or (**D**) BMS-200261 (PAR-1 inhibitor) at doses indicated and then stimulated with SSc-IgG (1 mg/mL) for 24 h (n = 5). To check for specificity of the effect of SSc-IgG, sister HMECs in D were treated with IgG from healthy controls (Con-IgG) at the same dose. For comparison, separate HMECs in D were treated with thrombin. ANOVA mean +/− SEM with * *p* < 0.05.

**Figure 3 ijms-22-11793-f003:**
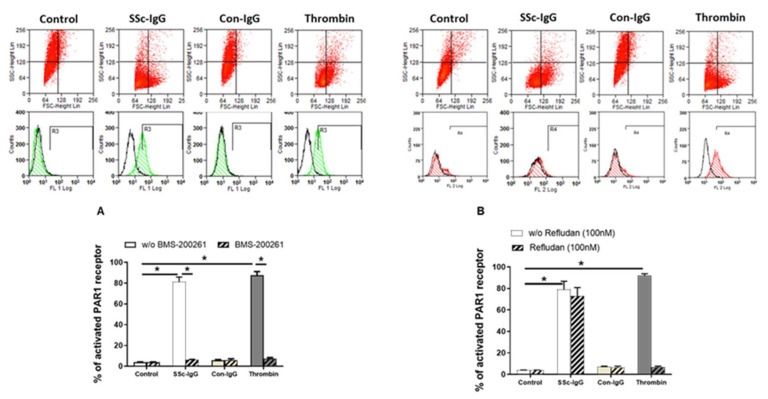
Effect of SSc-IgG on PAR-1 activation. HMECs were stimulated with SSc-IgG, Con-IgG or thrombin (0.1 U/mL) for 60 min and PAR-1 activation was then assessed by measuring the percentage of cleaved PAR-1 on the cell surface. HMECs were pretreated with either (**A**) BMS-200261 (PAR-1 inhibitor) or (**B**) refludan (thrombin inhibitor) for 24 h prior to stumulation. Representative histograms are from flow cytometry (FACS) with HMECs labelled with or without SPAN12 antibody, as described in Methods. The data on bar graphs are means ± SEM from 4 experiments with * *p* < 0.05.

**Figure 4 ijms-22-11793-f004:**
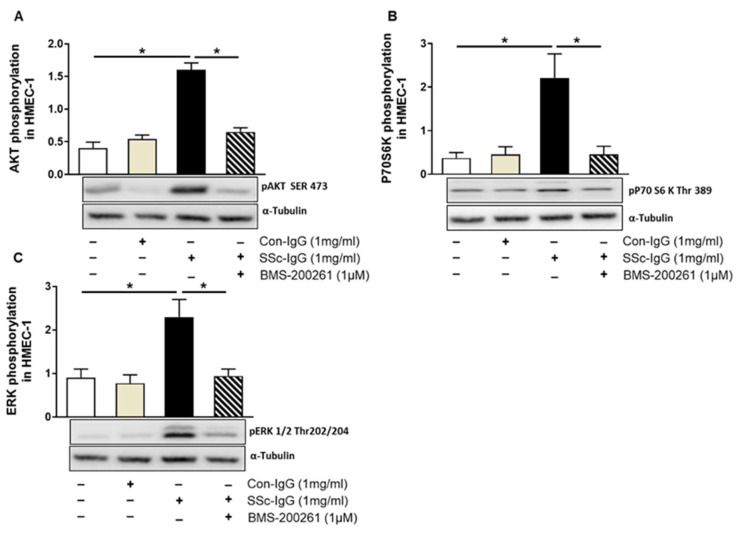
Effect of SSc-IgG on AKT, 70S6K and ERK1/2 phosphorylation. HMECs were pretreated with or without the PAR-1 inhibitor BMS-200261 for 1 h, followed by stimulation with either SSc-IgG or Con-IgG (both at 1 mg/mL) for 15 min and assessed for the presence of phosphorylated (**A**) AKT, (**B**) p70S6K or (**C**) ERK1/2 (n = 4). The data are expressed as the ratio of target protein expression to α-tubulin expression. ANOVA mean +/− SEM with * *p* < 0.05.

**Figure 5 ijms-22-11793-f005:**
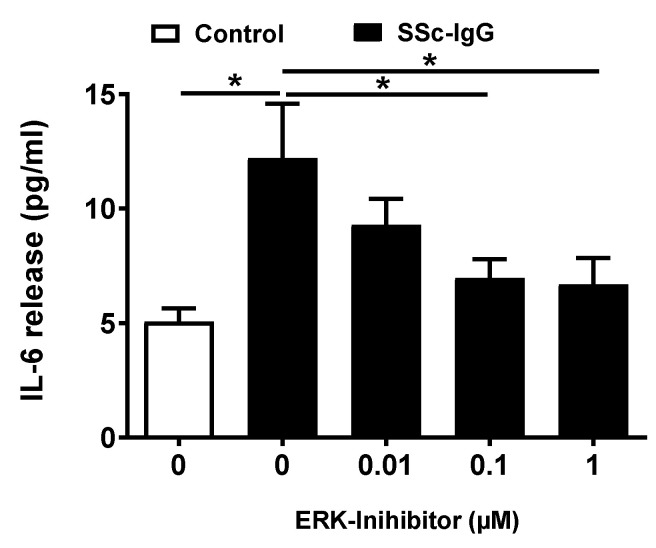
Effect of ERK1/2 signalling on SSc-IgG-induced IL-6 release by HMECs. Cells were pretreated with or without the specific ERK 1/2 inhibitor PD-184352 for 1 h, then stimulated with SSc-IgG (1 mg/mL) for 24 h after which the level of IL-6 released was measured (n = 8). ANOVA mean +/− SEM with * *p* < 0.05.

**Figure 6 ijms-22-11793-f006:**
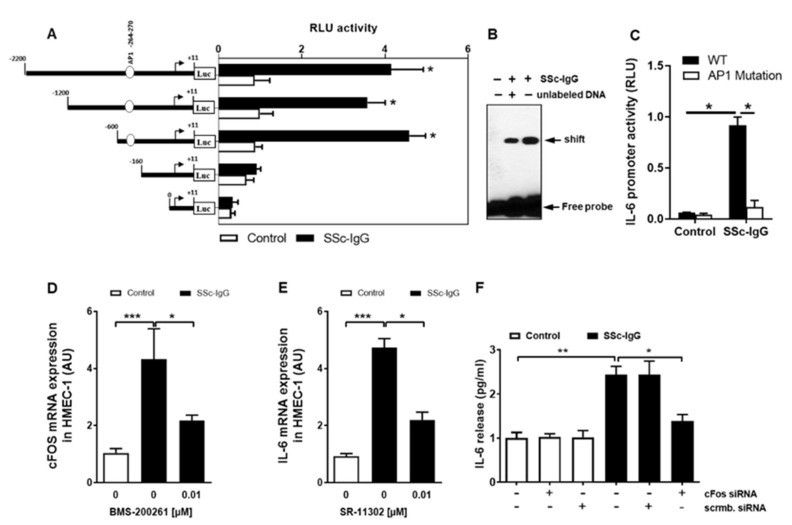
Characterization of the effect of SSc-IgG on the activity of the IL-6 promoter. HMECs were transiently transfected with IL-6 promoter constructs, stimulated with SSc-IgG (1 mg/mL) for 6 h and analyzed for luciferase activity. (**A**) Effect of progressive 5′ deletions of the IL-6 promoter on its activity upon stimulation with SSc-IgG (n = 8). (**B**) EMSA identifying the role of c-FOS in activation of the IL-6 promoter by SSc-IgG (n = 4). Nuclear extracts were obtained from HMECs treated with or without SSc-IgG (1 mg/mL) for 6 h, and EMSA was performed using c-FOS consensus oligonucleotide probes. (**C**) Effect of mutation in the AP-1 (c-FOS complex with cJUN) binding site within the IL-6 promoter on its activity after stimulation with SSc-IgG. HMECs were transfected with a normal or a mutated construct and stimulated with SSc-IgG (1 mg/mL) for 6 h (n = 4). (**D**) c-FOS mRNA expression by HMECs stimulated with SSc-IgG (1 mg/mL) for 1 h. Cells were pretreated with or without BMS-200261 for 1 h prior to stimulation (n = 4). (**E**) IL-6 mRNA expression by HMECs stimulated with SSc-IgG (1 mg/mL) for 3 h. Cells were pretreated with or without SR-11302 (AP-1 inhibitor) for 1 h prior to stimulation (n = 4). (**F**) IL-6 release by HMECs transfected with either c-FOS siRNA or control scrambled siRNA and stimulated with SSc-IgG (1 mg/mL) for 24 h (n = 4). ANOVA mean +/− SEM with * *p* < 0.05, ** *p* < 0.01, and *** *p* < 0.001.

## Data Availability

Original data are available upon Request.

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
