# Peer review of "Autoantibodies from Patients with Scleroderma Renal Crisis Promote PAR-1 Receptor Activation and IL-6 Production in Endothelial Cells"

_ijms, 2021, doi:10.3390/ijms222111793_

Round 1

Reviewer 1 Report

It is a really interesting study about the promotion of PAR-1 receptor activation and IL-6 production in endothelial cells by autoantibodies from patients with scleroderma renal crisis. It is very important that the results have implications for the treatment of these patients.

The design of the study is adequate and a lot of methods are used to prove the assumptions made by the authors. Because of the number of the experiments and methods used it would be necessary if the authors presented a figure or a table with the sequence of the steps - methods and conclusions, leading to the final conclusion. Some minor spelling errors should be checked and corrected. A figure including flow cytometry histograms, dot plots, showing PAR-1 expression and activation is needed.

Author Response

We thank the reviewer 1 for this positive assessment and have answered all the comments in point-by-point answers attached.

Reviewer 2 Report

Herein, the authors report that autoantibodies against PAR-1 receptor found in patients with systemic sclerosis (SSc) and SRC induce IL-6 production by endothelial cells. The mechanisms leading to this production involve  the signaling pathways controlled by the AP-1 transcription factor. These results are exciting and could, as suggested by the authors, open new perspectives of treatment. This work seems very original to me. However, I would like to moderate this thought, by reminding that AP-1 has been targeted by many researches for years (especially for fibrosis and. SSc). for scleroderma renal crisis (SRC), it seems very interesting and promising for translational approaches.

1. The role of AP1 in fibrosis should be better discussed and references should be added and discussed, as for examples: 

  • Murthy S, Wannick M, Eleftheriadis G, Müller A, Luo J, Busch H, Dalmann A, Riemekasten G, Sadik CD. Immunoglobulin G of systemic sclerosis patients programs a pro-inflammatory and profibrotic phenotype in monocyte-like THP-1 cells. Rheumatology (Oxford). 2021 Jun 18;60(6):3012-3022. doi: 10.1093/rheumatology/keaa747. PMID: 33230552.
  • Avouac J, Palumbo K, Tomcik M, Zerr P, Dees C, Horn A, Maurer B, Akhmetshina A, Beyer C, Sadowski A, Schneider H, Shiozawa S, Distler O, Schett G, Allanore Y, Distler JH. Inhibition of activator protein 1 signaling abrogates transforming growth factor β-mediated activation of fibroblasts and prevents experimental fibrosis. Arthritis Rheum. 2012 May;64(5):1642-52. doi: 10.1002/art.33501. PMID: 22139817.  

2. Fos-related antigen-2 (Fra-2) belongs to the activator protein-1 (AP-1) family of transcription factors. The Fra-2 transgenic model is a model of SSc. Have the authors evaluated the production of IL-6 in this model ? Please provide additional results (even and maybe in supplementary ) and/or dIscuss this model and this comment on IL6.

3. Maybe, healthy controls are  not the best controls. Can the authors provide some results with other samples from patients with connective tissue disorders other than SSc ?  

4. GPCR antibodies must be defined, as well as PAR, AT1-R, ETA-R and PAR-1. Please provide clear definitions.  In addition, positivity for these are not clearly given. Please improve. A table for patients ans samples would be useful.

5. a German word slipped into the text (line 109)

Author Response

We thank the reviewer 2 for this positive assessment and have answered all the comments in point-by-point answers attached.
